# A Multiblock Approach to Fuse Process and Near-Infrared Sensors for On-Line Prediction of Polymer Properties

**DOI:** 10.3390/s22041436

**Published:** 2022-02-13

**Authors:** Lorenzo Strani, Raffaele Vitale, Daniele Tanzilli, Francesco Bonacini, Andrea Perolo, Erik Mantovani, Angelo Ferrando, Marina Cocchi

**Affiliations:** 1Department of Chemical and Geological Sciences, University of Modena and Reggio Emilia, Via 4 Campi 103, 41125 Modena, Italy; lostrani@unimore.it (L.S.); daniele.tanzilli@unimore.it (D.T.); 2Centre National de la Recherche Scientifique (CNRS), Laboratoire de Spectroscopie pour les Interactions, la Réactivitè et l’Environnement (LASIRE), Cité Scientifique, University Lille, F-59000 Lille, France; raffaele.vitale@univ-lille.fr; 3Research Center, Versalis (ENI) S.p.A., Via Taliercio 14, 46100 Mantova, Italy; francesco.bonacini@versalis.eni.com (F.B.); andrea.perolo@versalis.eni.com (A.P.); erik.mantovani@versalis.eni.com (E.M.); angelo.ferrando@versalis.eni.com (A.F.)

**Keywords:** Acrylonitrile-Butadiene-Styrene, low-level data fusion, multiblock-partial least squares (MB-PLS), multivariate statistical process control, polymer production, quality prediction, real-time monitoring, response-oriented sequential alternation (ROSA)

## Abstract

Petrochemical companies aim at assessing final product quality in real time, in order to rapidly deal with possible plant faults and to reduce chemical wastes and staff effort resulting from the many laboratory analyses performed every day. In order to answer these needs, the main purpose of the current work is to explore the feasibility of multiblock regression methods to build real-time monitoring models for the prediction of two quality properties of Acrylonitrile-Butadiene-Styrene (ABS) by fusing near-infrared (NIR) and process sensors data. Data come from a production plant, which operates continuously, and where four NIR probes are installed on-line, in addition to standard process sensors. Multiblock-PLS (MB-PLS) and Response-Oriented Sequential Alternation (ROSA) methods were here utilized to assess which of such sensors and plant areas were the most relevant for the quality parameters prediction. Several prediction models were constructed exploiting measurements provided by sensors active at different ABS production process stages. Both methods provided good prediction performances and permitted identification of the most relevant data blocks for the quality parameters’ prediction. Moreover, models built without considering recordings from the final stage of the process yielded prediction errors comparable to those involving all available data blocks. Thus, in principle, allowing final ABS quality to be estimated in real-time before the end of the process itself.

## 1. Introduction

Nowadays, in several different domains like precision agriculture as well as pharmaceutical, food and chemical manufacturing, it is very common to utilize many analytical sensors to comprehensively characterize complex systems under study and to monitor processes while they evolve over time [1]. Analyzing the data yielded by such sensors by means of appropriate statistical tools is challenging but crucial in order to obtain meaningful physico-chemical information and design efficient production monitoring and control schemes. In particular, in industrial applications, a relevant issue is how to integrate or fuse the data resulting from sensors of different nature, potentially installed at different locations in the plant and in real time.

Multivariate Statistical Process Control (MSPC) is a well-established tool to accomplish real time monitoring and control of industrial production, in particular Latent Variables-Based MSPC (LV-MSPC) [2,3,4,5,6,7]. Most LV-MSPC relies on so-called engineering process variables [8], i.e., measured by on-line sensors controlling machinery settings (such as flow-meters, temperature and pressure probes, etc.) to build reference multivariate models for normal operating conditions (NOC), which are afterwards used to derive multivariate control charts and/or predicting quality attributes of finite product. More recently, thanks to technological developments, spectroscopic probes, especially near-infrared (NIR) ones, are extensively exploited [6,7,9,10,11,12,13] to monitor process evolution, or, in other words, to determine intermediate and final product quality parameters. Many studies in literature report on these aspects. Their results mainly refer to pilot scale plants [9,11,12,14] as well as to batch types of processes and seldom are engineering process variables and NIR measurements combined for constructing LV-MSPC models [6,10,14,15].

Fusing spectra with engineering variables is not a trivial task. However, process monitoring and control can greatly benefit from fusing these diverse data types, since, in this way, chemical composition-related information and physical and mechanical behavior/properties can be integrated.

This work focuses on a continuous styrenic polymer production process [16], monitored by means of NIR probes installed on-line in a production plant, as well as by standard process sensors. The main aim is to build real-time monitoring models to predict two of the main quality attributes of the final polymeric product by fusing NIR and process sensors’ data. A preliminary feasibility study was recently conducted by the authors at the pilot-plant level [14].

Two aspects are particularly relevant for industry: (i) the possibility of estimating in real time the quality of a finite product, thus reducing the operational time and the amount of chemicals commonly required for laboratory off-line assessments by reference methods; and (ii) to reach the anticipated assessment of departure from desired quality before the end of production itself, in order to plan possible early modifications of the operating settings.

To this end, we investigated the application of multiblock chemometric methods [17,18,19,20,21,22,23,24,25] which are suitable to accomplish data fusion at low-level [26,27] and might bring interesting advantages with respect to alternative mid-level and high-level data integration strategies [26] especially in terms of model training, maintenance and interpretability. In fact, original variables are directly used without any compression steps, and it is possible to assess the salience of each block/type of sensors in the model, i.e., inspecting their degree of uniqueness or redundancy.

In particular, we compared a well-established multiblock MSPC approach, such as MultiBlock Partial Least Squares (MB-PLS) regression [21], with Response-Oriented Sequential Alternation (ROSA) [22]. The distinctive features of ROSA, which is also based on PLS regression [28,29], are: (i) to be invariant to block scaling and not to be affected by the spurious bias resulting from the combination of data blocks of different size (similarly to sequential orthogonal PLS (SO-PLS) [20]); and (ii) to be computationally efficient and capable of dealing with any number of blocks, also a very high number (differently from SO-PLS).

We tested models constructed on measurements yielded by sensors that were active at all different process stages (up to the process production end), as well as models where measurements from the last stage were excluded. This was in order to evaluate if polymer quality could be forecasted prior to the end of production. The results achieved, by both MB-PLS and ROSA, show satisfactory predictive performance for the determination of the two quality parameters investigated. At the same time, the most relevant data blocks were assessed.

## 2. Materials and Methods

### 2.1. Process Description

Data presented in the current work were collected on-line in an Acrylonitrile-Styrene-Butadiene (ABS) industrial production plant (full scale) operating in continuous process, owned by Versalis (ENI group). For the sake of simplicity, the plant can be regarded as divided into five different areas: (i) pre-poly/mixer, where the three precursor monomers (acrylonitrile, styrene and butadiene) are mixed together; (ii) reaction point A; (iii) reaction point B; (iv) reaction point C; and (v) devolatilizer/cut zone, where the finite product is cut. Throughout all these areas seventy process sensors (PS), which measure temperatures, pressures, flow rates and motor speed, and four NIR probes are installed. The NIR probes are placed in four specific and crucial areas of the production plant: one where dissolution of butadiene in styrene occurs, before the addition of acrylonitrile; one in the pipe for the recovery of condensed reagents; one between the first and the second reaction points; and one at the very end of the process, just before the cut zone. Overall, both PS and NIR probes record data/spectra with a frequency of about one minute. In this study, data registered from January 2020 to May 2021 were analyzed, even if not all the data recorded during this period were considered in model building, due to production pauses and deviations from the operative conditions relevant for the current study.

### 2.2. Reference Analysis

Two different parameters have been considered for the evaluation of ABS quality. Nonetheless, because of confidential agreement restrictions with the company, their actual names will not be disclosed, but they will be referred to as Property 1 and Property 2. Properties 1 and 2 are assessed off-line by collecting ABS samples, i.e., final product, two (Property 1) and three (Property 2) times per day. Property 1 is related to ABS composition, i.e., the percentage of a certain chemical compound in the final product. On the other hand, Property 2 gives information about physical features of the product and the values of the related reference analysis are expressed in grams. In the period covered by this study 597 and 904 laboratory tests (homogeneously distributed all over the time period) were carried out to determine Property 1 and Property 2, respectively. Property 1 values ranged from 20 to 21.8%; Property 2 values ranged from 3.9 to 6.1 g.

### 2.3. NIR Spectroscopy

A Matrix FT-NIR spectrometer (Bruker Optics, Milan, Italy) was used to acquire spectra in the four different acquisition sites. The instrument was equipped with optical fibers (length: 100 m, diameter: 600 μm), whose probes (HT immersion probe, Drawing-no. 661.2350_1, Hellma GmbH and Co. KG, Müllheim, Germany) were directly connected to the four different acquisition sites located on the process pipe. Spectra were collected in transmission mode over the 12,500–4000 cm^−1^ spectral range, with a nominal resolution of 4 cm^−1^ (64 scans per sample).

### 2.4. Data Analysis

#### 2.4.1. Data Block and Multiblock Arrangement

The ensemble of collected data was arranged into nine distinct data blocks, according to the data type and the acquisition area along the process: on the one hand, PS measurements were gathered in five blocks, one per every area of the plant (see also Section 2.1); on the other hand, NIR spectra were arranged into four blocks, each corresponding to an individual optical probe. In Table 1, the names and abbreviations (which will be hereafter used) of all the blocks are shown, together with their size and the location along the plant. This is also an indication of how they are ranked in time, being a continuous process.

For both multiblock approaches, the data blocks were assembled considering the chronological progression of the ABS production process and, therefore, based on the location of the different sensors along the production line. In other words, each data point present in the datasets refers to information collected at different times, but it is correctly matched to the same processed material (i.e., data are synchronized).

Figure 1 displays a schematic representation of the low-level data fusion strategy adopted.

#### 2.4.2. Preprocessing


Individual block preprocessing


Prior to the multiblock modeling phase, each data set was preprocessed individually. In particular, variables in each PS data block were scaled to unit variance (different in nature and scales) whereas spectra, in each NIR data block, were baseline-corrected by using automatic weighted least squares [30]. Moreover, only the spectral range from 6500 to 5000 cm^−1^ (the sole one exhibiting spectral bands ascribable to either reactants or products) was taken into account for subsequent model training. Figure 2 shows the effect of the baseline correction executed on the NIR spectra of the NIR-RP-A data block.


Multiblock preprocessing


After the individual preprocessing of the single blocks, each data set was scaled to unit block variance (including column mean-centering) prior to MB-PLS [21]. In fact, MB-PLS operates directly on row-wise concatenated data blocks and a fair block contribution has to be assured.

Concerning ROSA, the individual pre-processed blocks were just mean-centered since such a method treats one block at a time, as it will be detailed in the following sections.

#### 2.4.3. MB-PLS

We exploited here the MB-PLS implementation originally proposed by Westerhuis and Coenegracht [31] which can be looked at as standard PLS with appropriate block scaling steps as described in [21]. Thus, MB-PLS is an extension of the classical PLS regression [28] for applications involving different data blocks that share the same number of rows (observations), relating to the data matrix **X**, resulting from the row-wise concatenation of *N* different data blocks (Equation (1)):**X** = [**X**_1_, **X**_2_, …, **X***_N_*](1)
to the response(s) of interest.

This method provides global (also called *super-*) scores, weights, loadings and regression coefficients, as well as local (also called *block-*) scores and weights for each data block, as it is shown in Equations (2)–(5):**w***_b_* = **X^T^***_b_* ∗ **u**/**u^T^u**(2)
**t***_b_* = (**X***_b_* ∗ **w***_b_*)/√**n***_b_*(3)
**w** = **T^T^** ∗ **u**/**u^T^u**(4)
**t** = **T** ∗ **w**/**w^T^w**(5)
where **n***_b_* is the number of variables in a given block, **t***_b_* and **w***_b_* are the local scores and weights, respectively, whereas **t** and **w** are the global (*super*) scores and weights. **T** is yielded by the concatenation of all **t***_b_*.

This way, it is possible to assess the contribution of each data block (analyzing **w***_b_* for the prediction of the response variable/s **y**/**Y**, improving the process understanding).

#### 2.4.4. ROSA

Response-Oriented Sequential Alternation (ROSA) is a multiblock regression method proposed by Liland et al. [22] that is also based on PLS regression. Different from MB-PLS, in that ROSA is a sequential algorithm, similar to, e.g., SO-PLS [20], which renders the method invariant with respect to block-scaling (blocks are just mean centered), as well as to block ordering, differently from SO-PLS. These features allow dealing with a large number of blocks of different dimensions.

Moreover, ROSA exhibits a high computational efficiency, as it does not require the iterative convergence of an optimization criterion, and because only the response is deflated, not all the blocks. In fact, each PLS component is selected from a single block, picking among the various covariance-maximizing candidate components, estimated from each data block, the one returning the smallest prediction residuals. Successive components are constrained to be orthogonal to the subspace spanned by the previously winning components. Thus, scores’ and loadings’ orthogonality is ensured.

The ROSA algorithm for a single response variable, **y**, is summarized in the following equations:**w***_b_* = **X***_b_*^T^ ∗ **y**(6)
**t***_b_* = **X***_b_* ∗ **w***_b_*(7)
**r***_b_* = **y** − **t***_b_* **t***_b_*^T^**y**(8)
where **X***_b_* is a single data block, while **w***_b_*, **t***_b_* and **r***_b_* are block weights, scores and residuals, respectively. The first component is selected as the one computed from the *b_th_*-block yielding the smallest residuals (**r***_b_*), and **t**_1_ are taken to be equal to **t***_b_* of the winning block. The corresponding weights and scores are normalized (and also orthogonalized with respect to the preceding components from the second component on). The **y**-loadings are finally estimated as:**q***_a_* = **y**^T^ **t***_a_*(9)
where **t***_a_* are the scores previously selected for the *a_th_* LV.

**X**-loadings (**P**) and PLS regression coefficients (**b**) (and possibly a constant term b_0_) can be estimated according to the Equations (10)–(12), after selecting the number of optimal LVs and collecting the corresponding scores, weights, **y**-loadings in matrix array **T**, **W** and **Q**.
**P** = **X**^T^ **T**(10)
**b** = **W**(**P**^T^**W**)^−1^**Q**(11)
**b**_0_ = **y**_m_ − **x**_m_ ∗ **b**(12)
where **y**_m_ is the mean of **y** and **x**_m_ is a vector with the mean for each variable of **X**.

Thus, each selected LV in ROSA encodes information proceeding only from the winning *b_th_*-block (the one achieving smallest residuals according to Equation (8)), and all LVs are orthogonal. It is important to notice that all blocks are always candidates at each algorithmic step. Therefore, consecutive LVs can depict information from the same block previously selected, or from a different one.

#### 2.4.5. Multiblock Models Building

With the aim of developing predictive models for the two parameters taken into account in this study and assessing which are the most important data blocks for their estimation, both MB-PLS and ROSA were investigated.

All the available data were split into calibration and validation sets for both Property 1 and Property 2. In order to assess models’ performance in a scenario mimicking a real-time application, the calibration sets comprised data collected during the year 2020 (~70% of total data), whereas the validation sets comprised data collected in 2021. Clearly, only samples, i.e., time points, for which the offline reference measurement were available were taken into account.

The two optimized best-performing models were finally utilized for assessing the values of Property 1 and 2 at time points where no reference data were acquired, in order to check whether the resulting estimations spanned a similar properties values range with respect to close time points.

In order to establish the complexity, i.e., number of PLS components, of each model, venetian blinds cross-validation with ten cancellation groups for Property 1 and four cancellation groups for Property 2 was resorted to. Model reliability was determined in terms of both root mean square error in cross-validation (RMSECV) and root mean square error in prediction (RMSEP).

Data blocks were preprocessed as described in Section 2.4.2.

For both MB-PLS and ROSA, the contribution of each block and block variables in the final predictive model was assessed by investigating the PLS regression coefficients and Variable Importance in Prediction (VIP) [32,33]. PLS block-weights were also inspected but, for the sake of brevity, the related figures are not reported, as the provided information was similar to that obtained by regression coefficients.

### 2.5. Software

All the chemometric analyses were performed using routines and toolboxes implemented in the MATLAB environment (the Mathworks Inc., Natick, MA, USA).

MB-PLS has been calculated through the PLS-Toolbox version 8.9 (Eigenvector Research Inc., Wenatchee, WA, United States).

ROSA (with options for venetian blind cross-validation, VIP calculation and validation sample response prediction) was implemented by the authors based on the MATLAB code provided in ref. [22].

## 3. Results

### 3.1. Property 1 Prediction

When all the available data blocks (PS and NIR measurements for all plant areas) were simultaneously modelled ROSA resulted to be the most performant method for the prediction of Property 1, yielding a RMSEP of 0.14%. On the other hand, MB-PLS returned a RMSEP value of 0.2%. This difference, however, is not substantial. The results are shown in Table 2 and Figure 3. ROSA selected only three of the nine blocks under study, two of which, Devo/cut and NIR-cut, relate to the last stage of the process, where the polymerization is over and the product is ready to be cut. Furthermore, among the 13 latent variables selected through the cross-validation procedure (aimed at minimizing RMSECV), eight were calculated from the NIR-cut block, which highlights a crucial relevance of the final NIR sensor, in this case, for the quality prediction. Figure 3a shows how the predictions for the objects of the validation set are homogeneously distributed within the expected range of the quality parameter concerned. In Figure 3b–d the PLS regression coefficients associated to the three blocks selected by ROSA are represented (the red stars denote variables/spectral regions whose VIP scores were higher than one). In the RP-A data block (selected only one time out of 13) only three temperature sensors were found relevant for Property 1 prediction, whereas in Devo/cut and NIR-cut data blocks all the sensors and nearly all the spectral regions sampled were somewhat important. In Figure 3d it is evident that the largest (in absolute value) regression coefficients are those corresponding to bands centered at 5900 cm^−1^ and 5250 cm^−1^ that can be ascribed to the investigated ABS compound.

Although such results might already be considered relatively satisfactory from a predictive point of view, two additional aspects would be worth investigating: (i) whether reasonably good quality prediction of Property 1 values could be obtained before the product is cut (i.e., without relying on sensors installed within the cut area); and (ii) whether the exclusive use of spectral sensors or process sensors could be sufficient for a reliable estimation of this quality index. To this end, in addition to the dataset containing all the blocks, MB-PLS and ROSA models were calculated using fused datasets comprising only the blocks before the cut zone, only PS data and only NIR data (both including and excluding the spectra contained in the NIR-cut block), respectively.

Table 2 reports the results of all the computed multiblock prediction models related to Property 1. It is possible to observe that prediction errors resulting from ROSA are systematically lower than the one obtained by means of MB-PLS. It is also clear how NIR data are far more important for the prediction of Property 1 than PS data. In fact, when ROSA is run on both block types, components from NIR data sets are more often selected than those computed from PS data sets. Moreover, in MB-PLS models, variables related to NIR blocks are always relevant for Property 1 prediction. In addition, the RMSEP of models that are calculated using only NIR data is comparable to that of models using both PS and NIR data, while using only PS data blocks results in a significant increase of the prediction error in calibration, cross-validation and external validation. This is somehow expected, as Property 1 is linked to ABS chemical composition and, therefore, an analytical technique like NIR spectroscopy is definitely more suitable for its determination than more standard engineering PS probes, which only indirectly reflect how fluctuations in the process operating conditions may affect the polymer characteristics.

Since ROSA models always selected components estimated from the blocks located on the plant cut area, i.e., blocks eight and nine, we also decided to calibrate ROSA models (using both PS and NIR data and only NIR data) excluding completely such blocks from the computational procedure (see ‘ROSA no cut zone’ and ‘ROSA only NIR no cut zone’ in Table 2, respectively). In both cases, RMSEP values for models not including the cut area, were found higher, yet acceptable by process operators. This clearly makes it possible to retrieve reasonable Property 1 value estimate before the completion of the ABS production process. Moreover, similar prediction errors were obtained by using only NIR blocks or when combining NIR and PS blocks. Hence, two possible pathways can be envisioned for the real-time prediction and control of Property 1: (i) resorting to both data types and getting a clearer idea of the important process areas/sensors all along the production plant; or (ii) just exploiting NIR spectra for more efficient data management and to deal with less noisy data.

In order to evaluate the role of all types of sensors, Figure 4 displays the results yielded by the ‘ROSA no cut zone’ model. It is worth mentioning that half of the blocks selected by the ROSA algorithm relate to the reaction points A and B, whereas the other half to the NIR-cond data block, whose respective probe is right before these reaction points. Looking at the order (not reported for the sake of brevity) in which blocks were selected by ROSA, it can be observed how the winning blocks for the first five latent variables were RP-B (picked only one time) and NIR-RP-A (picked four times). For the remaining model dimensions, NIR-cond was selected six times in a row, while RP-A and RP-B one each. Details about the selection order are useful to assess which blocks, i.e., areas of the plant, encode the most important information for the prediction of the investigated quality parameter.

Figure 4b,d show the regression coefficients for the two aforementioned NIR blocks, with NIR-RP-A exhibiting a larger number of spectral variables characterized by VIP scores higher than one, especially in the region between 5400 cm^−1^ and 5250 cm^−1^, that are ascribable to the stretching of a functional group of one of the three precursor compounds on which Property 1 directly depends. Conversely, in Figure 4c,e the regression coefficients for the PS data blocks are graphed: the most significant variables, according to their respective VIP values, are almost all related to temperature and motor speed sensors installed in different subzones of the reaction points A and B.

### 3.2. Property 2 Prediction

The same model building strategy described before was finally followed for the prediction of Property 2. Table 3 reports the results obtained by means of both MB-PLS and ROSA. ROSA, when all the available data blocks were simultaneously modelled, did not select any cut area block, therefore the ‘ROSA no cut zone’ model was not trained in this case.

MB-PLS models calibrated by using (i) all the data blocks or (ii) only PS data returned the most satisfactory results, contrary to the results obtained for Property 1. In fact, the influence NIR spectra have on the estimation of Property 2 prediction is not predominant, except for the NIR-RP-A block, which was selected many times by the ROSA algorithm and whose variables always showed VIP scores higher than one in MB-PLS. These results can be interpreted in the light of the fact that Property 2 is not linked to the chemical composition of ABS but evaluates the performance of the finite product as determined by mechanical/physical tests. Subsequently, it is undoubtedly more affected by variability occurring in the processing steps, and can change significantly even if the aforementioned chemical composition does not change. RMSEP increased up to 0.52 g when no PS block was considered. However, for models built without PS data, MB-PLS achieved a slightly better performance than ROSA (0.48–0.5 g vs. 0.52 g). These results suggested how the exclusive use of NIR sensors is not sufficient for a reliable estimation of Property 2.

Overall, MB-PLS showed a better prediction performance for Property 2. The best results were obtained by the ‘MB PLS all’ model (RMSEP = 0.34 g), even though ‘MB PLS no cut zone’ and ‘MB PLS only PS’ provided similar results.

In Figure 5 is where the predicted vs. measured value plot resulting from the ‘MB-PLS all’ model is shown. By inspecting the corresponding residuals plot (not shown for the sake of brevity) it can be observed that, on average, the 2021 production campaign (validation set), yielded lower values of Property 2 than that conducted in 2020 (calibration set). This deviation explains the relatively high difference between RMSEP and RMSEC and RMSECV. However, the presence of a reasonable amount of validation samples in the whole calibration range was guaranteed and the company deemed the prediction error acceptable for routine monitoring.

In Figure 6 the ‘MB-PLS all’ model regression coefficients are reported. All PS were found to be important for the prediction of Property 2 based on their VIP scores values. For what concerns the NIR blocks regression coefficients, the NIR-RP-A is confirmed to be the block with the largest number of highly predictive spectral regions, which are mainly related to the three precursors monomers of ABS. For the other NIR blocks, relevant regions of interest were found in correspondence of the absorption bands centered at 5900 cm^−1^ and 6100 cm^−1^, respectively.

### 3.3. Real-Time Predictions

Finally, Figure 7 illustrates the predicted values of Property 1 obtained through the ROSA model constructed on all data blocks (Table 2, row 1) for the time points for which reference response measurements were not acquired.

These predicted values span a range very similar to that covered within both the calibration and the validation set. A few slight deviations were observed, interestingly right after specific shut-down time periods: such deviations may, in fact, arise from the fact that many industrial processes (including polymerization processes) take a certain time to readapt to NOC conditions after particular external interventions (e.g., cleaning, maintenance, etc.).

Similar results were obtained for real-time predictions with the model ‘MB-PLS no cut zone’ for Property 2, as shown in Figure 8.

## 4. Conclusions

This work demonstrated how multiblock approaches could be used for the construction of reliable and robust real-time monitoring models for the on-line prediction of industrial quality parameters of ABS. In fact, the data partition in different blocks and the low-level data fusion strategy adopted here permitted to improve ABS production process understanding, enabling the assessment of the most crucial plant areas and the relevant sensors for the prediction of such specific parameters. Moreover, the application of these approaches is essential when two or more different analytical platforms of different nature, like the NIR spectrometer and more standard engineering process sensors, are simultaneously used to control any generic production process.

More specifically, in this article, both MB-PLS and ROSA allowed performant predictive models to be constructed for the two properties under study (i.e., Property 1 and 2). In particular, for the prediction of Property 1, ROSA resulted in a lower RMSEP compared to MB-PLS, highlighting the importance of NIR data over process sensor data when a chemical composition-related quality index is to be estimated. On the other hand, Property 2 was more efficiently predicted by a MB-PLS method, which pointed out a higher relevance of process sensors compared to NIR data when, instead, physical features need to be assessed.

Furthermore, models computed without taking into account measurements related to the final area of the plant (cut zone) provided comparable prediction errors with respect to the best models built on all the ensemble of available data. This is of great industrial interest, since, in principle, ABS quality could be determined before its production is completed, which might allow possible modifications of the plant settings and/or changes in the operating conditions to be planned in advance and with reduced costs.

In conclusion, these approaches could help in: (i) accelerating decision making and troubleshooting; (ii) reducing the amount of chemical waste generated in full-scale plants; (iii) decreasing the number of off-line laboratory tests required for quality control; and (iv) facilitating any type of operation along the production line as well as possible fault detection and diagnosis.

## Figures and Tables

**Figure 1 sensors-22-01436-f001:**
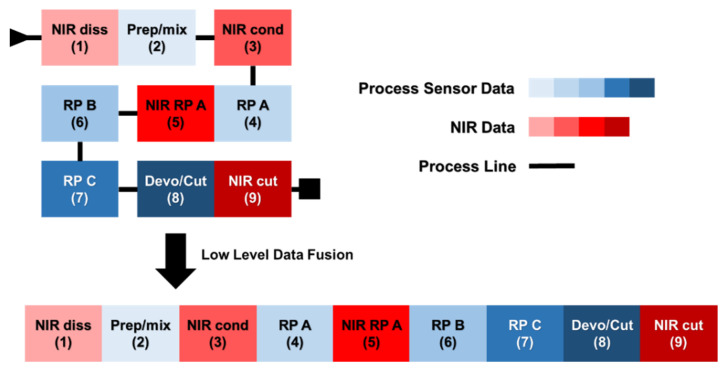
Schematic representation of the low-level data fusion approach resorted to in this study. Values in brackets indicate the chronological order of the data blocks.

**Figure 2 sensors-22-01436-f002:**
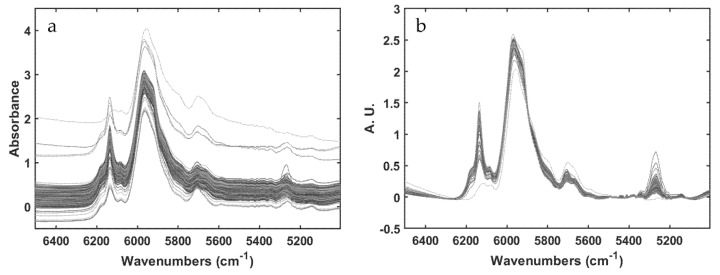
Spectra collected at NIR-RP-A, data block before (**a**) and after (**b**) baseline correction using automatic weighted least square method.

**Figure 3 sensors-22-01436-f003:**
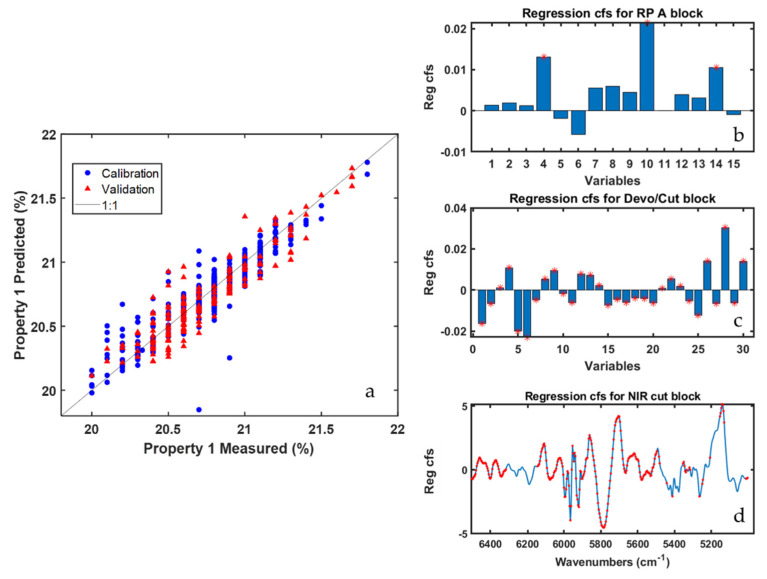
ROSA results for Property 1 prediction (all data blocks were modelled simultaneously). Predicted vs. measured value plot (**a**); regression coefficients for the RP-A (**b**); Devo/cut (**c**); and NIR cut (**d**) data blocks. Red stars indicate variables having VIP scores higher than one.

**Figure 4 sensors-22-01436-f004:**
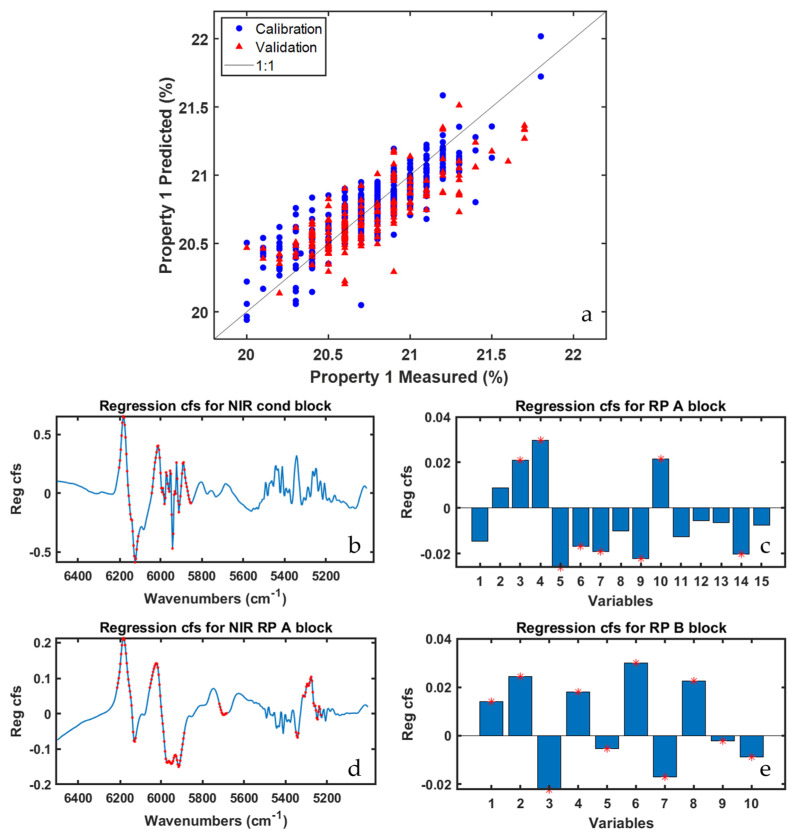
ROSA results for Property 1 prediction (‘ROSA no cut zone’ model). Predicted vs. measured value plot (**a**); regression coefficient for NIR cond (**b**); RP A (**c**); NIR RP A (**d**); and RP B (**e**) data blocks. Red stars indicate variables having VIP scores higher than one.

**Figure 5 sensors-22-01436-f005:**
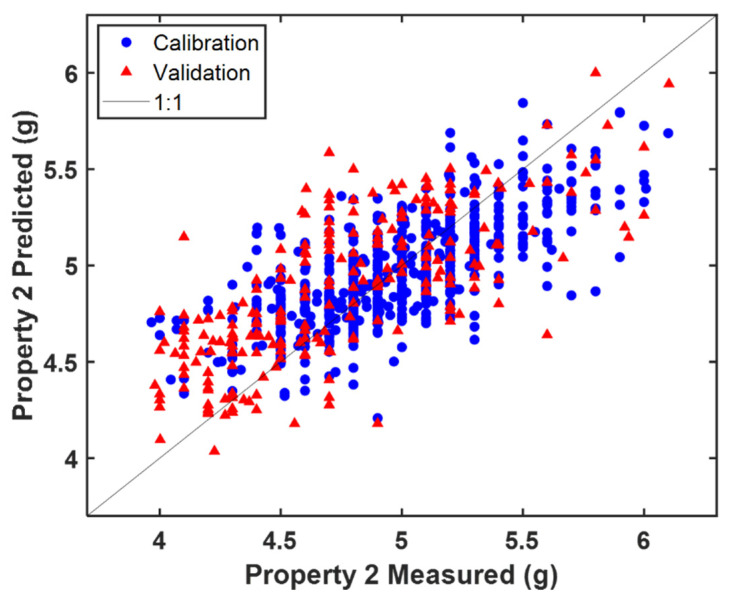
Predicted vs. measured value plot resulting from the ‘MB-PLS all’ model.

**Figure 6 sensors-22-01436-f006:**
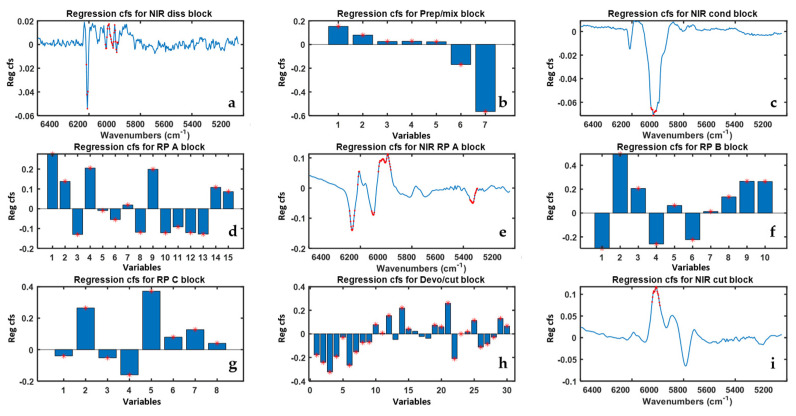
Regression coefficients resulting from the ‘MB-PLS all’ model for each data block the letters (**a**–**i**) refer to the different block whose name is reported on top. Red stars indicate variables exhibiting VIP scores higher than one.

**Figure 7 sensors-22-01436-f007:**
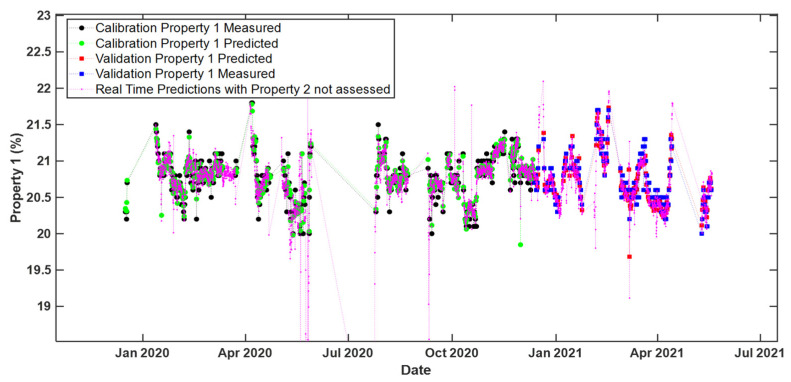
Real time predictions of Property 1 (i.e., time evolution of the measured and predicted values). The predictions were obtained by means of the ‘ROSA all’ model. Legend: black circles—calibration set measured values; green circles—calibration set predicted values; blue squares—validation set measured values; red squares—validation set predicted values; magenta dots—predicted values related to time points for which no reference response measurements were available. For ease of visualization only every 2 h predictions during the considered time period are shown.

**Figure 8 sensors-22-01436-f008:**
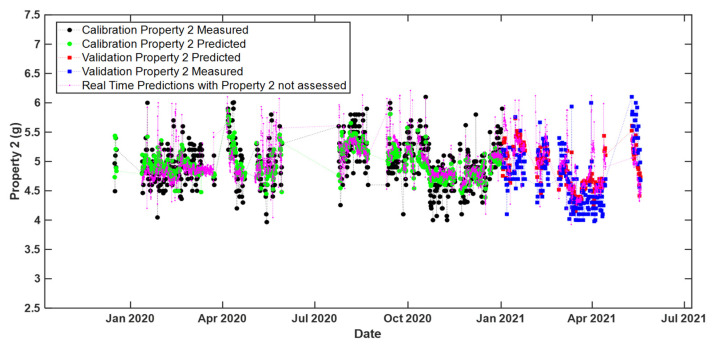
Real time predictions of Property 2 (i.e., time evolution of the measured and predicted values). The predictions were obtained by means of the ‘MB PLS no cut zone’ model. Legend: black circles—calibration set measured values; green circles—calibration set predicted values; blue squares—validation set measured values; red squares—validation set predicted values; magenta dots—predicted values related to time points for which no reference response measurements were available. For ease of visualization only every 2 h predictions during the considered time period are shown.

**Table 1 sensors-22-01436-t001:** Data block description.

Block Full Name	Block Abbreviated Name	Data Type	No. of Variables ^1^	Order
NIR dissolution	NIR-diss	NIR Spectra	390	1
Prepoli/Mixer	Prep/mix	PS	7	2
NIR condensation	NIR-cond	NIR Spectra	390	3
Reaction Point A	RP-A	PS	15	4
NIR Reaction Point A	NIR-RP-A	NIR Spectra	390	5
Reaction Point B	RP-B	PS	10	6
Reaction Point C	RP-C	PS	8	7
Devolatilizer/cut zone	Devo/cut	PS	30	8
NIR cut zone	NIR-cut	NIR Spectra	390	9

^1^ For NIR data blocks, the number of variables is equal to the spectra wave numbers, whereas for PS data blocks it is equal to the number of PS present in the respective plant area. The column “Order” highlights how the process evolves chronologically.

**Table 2 sensors-22-01436-t002:** Results yielded by MB-PLS and ROSA for the prediction of Property 1.

Model ID	Blocks Entering the Model	LVs	RMSEC (%)	RMSECV (%)	RMSEP (%)
MB PLS all	All	11	0.12	0.16	0.20
MB PLS no cut zone	1 to 7	11	0.13	0.17	0.23
MB PLS only PS	2–4–6–7–8	11	0.24	0.26	0.38
MB PLS only NIR	1–3–5–9	10	0.13	0.15	0.22
MB PLS only NIR no cut zone	1–3–5	8	0.14	0.15	0.22
ROSA all ^1^	4(1)–8(4)–9(8)	13	0.11	0.14	0.13
ROSA no cut zone	3(6)–4(1)–5(3)–6(2)	12	0.15	0.18	0.2
ROSA only PS	2(1)–4(6)–7(3)	10	0.23	0.25	0.31
ROSA only NIR	9(8)	8	0.12	0.13	0.14
ROSA only NIR no cut zone	3(12)–5(2)	14	0.16	0.18	0.19

^1^ the values in brackets indicate the number of times a certain block was selected by the ROSA algorithm.

**Table 3 sensors-22-01436-t003:** Results yielded by MB-PLS and ROSA for the prediction of Property 2.

Model ID	Blocks Entering the Model	LVs	RMSEC (g)	RMSECV (g)	RMSEP (g)
MB PLS all	All	10	0.25	0.27	0.34
MB PLS no cut zone	1 to 7	8	0.27	0.29	0.37
MB PLS only PS	2–4–6–7–8	9	0.27	0.29	0.35
MB PLS only NIR	1–3–5–9	7	0.34	0.34	0.48
MB PLS only NIR no cut zone	1–3–5	6	0.36	0.37	0.5
ROSA all ^1^	2(1)–4(1)–5(1)–6(1)	4	0.32	0.33	0.46
ROSA only PS	2(1)–4(1)–6(1)	3	0.32	0.33	0.45
ROSA only NIR	5(6)–9(3)	9	0.33	0.34	0.52
ROSA only NIR no cut zone	5(8)	8	0.33	0.34	0.52

^1^ The values in brackets indicate the number of times a certain block was selected by the ROSA algorithm.

## Data Availability

The datasets presented in this article are not readily available because of confidential agreement restrictions with the company. Requests to access the datasets should be directed to erik.mantovani@versalis.eni.com.

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
