# Peer review of "A Multiblock Approach to Fuse Process and Near-Infrared Sensors for On-Line Prediction of Polymer Properties"

_sensors, 2022, doi:10.3390/s22041436_

Round 1

Reviewer 1 Report

This paper tested the models including sensors from all stages of the process up to the final product and demonstrated how multiblock approaches could bring advantages for the online prediction of quality parameters of polymers. Both MB-PLS and ROSA methods achieved good prediction models for the two properties investigated. I think this article can be published in Sensors after modifying some of the expression problems.

Author Response

The authors thank the Reviewer for the positive feedback. The English has been deeply revised in order to make the text clearer and more understandable (the manuscript with changes highlighted in red, shows the text revisions).

Reviewer 2 Report

The subject is interesting. I am not quite sure how relevant can be to this journal, unless the authors add some introductory material to make the connection with the subject of “sensors”. Conclusions should also include what this research has to offer to scientists/professionals engaging in the area of sensors. Based on what has been written – the majority of references attest to this – the key contribution of this work could be more familiar to specialists in chemometrics and so forth.

Even so, it could attract a wider audience that might even involve researchers from information sciences and statistics.

The script is fairly well organized and presented.

The calibration-validation plots (Figs 3a and 4a) show an efficient approach to predict. The last comment is further supported by the MB-PLS and ROSA predictive performance as quantified by the RMSEC, RMSECV and RMSEP values for Property 1 and 2 (Tables 2 & 3). In Figure 5, the authors could provide regression fits to quantify the extent of overlap of between MB-PLS and VIP.

Author Response

The authors thank the Reviewer for the interest in our work and for pointing out possible weak points, which we addressed in the revision and are detailed in the following. 

 Remark 1

The subject is interesting. I am not quite sure how relevant can be to this journal, unless the authors add some introductory material to make the connection with the subject of “sensors”. Conclusions should also include what this research has to offer to scientists/professionals engaging in the area of sensors. Based on what has been written – the majority of references attest to this – the key contribution of this work could be more familiar to specialists in chemometrics and so forth.

The Authors are convinced that submitted work match the Special Issue aims, in particular the topic "Sensor data fusion analysis in Industrial Applications;"

Nonetheless, we follow the suggestion of making the connection clearer and added, in this respect, the two sentences reported below in the Introduction:

page 1, first paragraph:

 " Nowadays, in several different domains like precision agriculture as well as pharmaceutical, food and chemical manufacturing, it is very common to utilize many analytical sensors to comprehensively characterize complex systems under study and to monitor processes while they evolve over time [1]. Analyzing the data yielded by such sensors by means of appropriate statistical tools is challenging but crucial in order to obtain meaningful physico-chemical information and design efficient production monitoring and control schemes. In particular, in industrial applications, a relevant issue is how to integrate or fuse the data resulting from sensors of different nature, potentially installed at different location of the plant and in real time."

page 2, second paragraph:

" Fusing spectra with engineering variables is not a trivial task. On the other hand, process monitoring and control can greatly benefit from fusing these diverse data types, since, in this way, chemical composition-related information and physical, mechanical behaviour/properties can be integrated"

Moreover, refs. 1 and 8 are inserted, these are review articles which may be of interest to researchers in the “sensors” field

[1] Bowler, A.L.; Bakalis, S.; Watson, N.J. A review of in-line and on-line measurement techniques to monitor industrial mixing processes. Chemical Engineering Research and Design 2020, 153, 463-495

[8] Kadlec, P.; Gabrys, B.; Strandt, S. Data-driven Soft Sensors in the process industry. Computers and Chemical Engineering, 2009, 33, 795–814.

As concern the Conclusions, we are convinced that in the revised version of the manuscript the potential interest of our approach and results in the industrial applications of sensors are now better put in evidence, see e.g. the sentences at page 14 of Conclusions (2nd sentence of 2nd paragraph)

In particular, for the prediction of Property 1, ROSA resulted in a lower RMSEP compared to MB-PLS and, highlighted the importance of NIR data over process sensor data, when a chemical composition-related quality index is to be estimated. On the other hand, Property 2 was more efficiently predicted by a MB-PLS method, which pointed out a higher relevance of process sensors compared to NIR data, when, instead, physical features need to be assessed.

(page 14, 3rd paragraph)

Furthermore, models computed without taking into account measurements related to the final area of the plant (cut zone) provided comparable prediction errors with respect to the best models (either obtained with MB-PLS or ROSA) built on all the ensemble of available data. This is of great industrial interest, since, in principle, ABS quality could be determined before its production is completed, which might allow possible modifications of the plant settings and/or changes in the operating conditions to be planned in advance and with reduced costs. 

Remark 2

The calibration-validation plots (Figs 3a and 4a) show an efficient approach to predict. The last comment is further supported by the MB-PLS and ROSA predictive performance as quantified by the RMSEC, RMSECV and RMSEP values for Property 1 and 2 (Tables 2 & 3). In Figure 5, the authors could provide regression fits to quantify the extent of overlap of between MB-PLS and VIP.

The authors apologise to the reviewer, actually the second part of the figure 5 caption was not correct and this could have confused the reader. In fact, as highlighted by the legend, red triangles refers to validation samples. The comparison between calibration and validation results is reported in Table 3, whereas variables exhibiting VIP scores higher than one are shown in figure 6. Anyhow, we follo the suggestion to show the fit line in the plots, and Line 1:1 was added to figures 3a, 4a and 5, respectively.

Reviewer 3 Report

                                                 Journal of Sensors

The article entitled Multiblock approach to fuse process and NIR sensors for on-line prediction of polymer properties”. The article discusses that the main purpose of the current work was to use multiblock regression methods to build real-time monitoring models for the prediction of two quality properties of Acrylonitrile-Butadiene-Styrene (ABS) by fusing NIR and process sensors data. Multiblock-PLS (MB-PLS) and Response-Oriented Sequential Alternation (ROSA) methods were used to assess the most important plant areas for the quality parameters prediction. indicating the most significant data blocks/variables for the quality parameters prediction.

There are many technical, grammatical, and common mistakes in the article...........

          Technical, grammatical, and common mistakes are as follows

Comments

  • Write the full abbreviation of NIR in the title.
  • Write keywords in alphabetical order.
  • Keywords: Remove the abbreviation or write their full names.
  • Abstract: what is RMSEP? 1stly write their full name then their short abbreviation.
  • For %, °C, wt % and for Figure, etc., follow the same format throughout the manuscript.
  • Put the references of the manuscript through EndNote software.
  • Do not put more than three references for a single explanation throughout the manuscript.
  • Make the line and border of all figures bold.
  • Figure 2. Schematic representation. Make it step by step with the number given. The present format is complicated and not clear understandable for readers.
  • 4.3. ROSA is multiblock. It will be better if write their full name before starting a paragraph.
  • Section 2.4.4. To assess the reliability of prediction models, both root mean square error in cross-validation (RMSECV) and root mean square error in prediction (RMSEP) were taken into account. For which regards Multiblock-PLS models, data blocks were first preprocessed as described in 2.4.1 and then block scaled to unit block variance, in order to prevent that a single data block contributes more than the others just for containing a greater number of variables (as it is intrinsic to MB-PLS method). [Revise it].
  • Results. Figure 3. Explain it in more detail and compare it with some latest research results.
  • Section 3. Results. RMSEP increases up to 0.52 g when no PS block is considered. However, for models built without PS data. [ Why the authors considered] make it clear
  • The result part is very lengthy in reading feel boring. Please mention some heading. There should be individual heading for tables and graphs.
  • Figure 8. Draw it with high resolution.
  • (not shown for the sak eof brevity) correct the spelling.
  • Revise the conclusion and make it short.
  • Cite the following references;
  • https://doi.org/10.1007/s13726-020-00849-x
  • 2018, 6, (7):8856-8867. DOI: 10.1021/acssuschemeng.8b01212
  • 2020, 235 (10):1247-1262. DOI: https://doi.org/10.1515/zpch-2020-1697
  • 138 (14): 50138. https://doi.org/10.1002/app.50138
  • 2019, (38):21577-604. DOI: 10.1039/c9ta04575a
  • 2021; e50515. 1-10. https://doi.org/10.1002/app.50515

Warmly reminded; The present reference is not enough for this manuscript. The authors should want to add the latest and related references, besides the mentioned ones.

Reviewer 4 Report

The manuscript entitled: ‘Multiblock approach to fuse process and NIR sensors for on-line prediction of polymer properties’, is interesting and scientifically relevant. In this article, the analysis of using multiblock regression methods to build real-time monitoring models for the prediction of two quality properties of Acrylonitrile-Butadiene-Styrene (ABS) by fusing NIR and process sensors data, was carried out. Searching for new tools for acquiring and analyzing data in real time is important and therefore I recommend the work for publication.

Author Response

The manuscript entitled: ‘Multiblock approach to fuse process and NIR sensors for on-line prediction of polymer properties’, is interesting and scientifically relevant. In this article, the analysis of using multiblock regression methods to build real-time monitoring models for the prediction of two quality properties of Acrylonitrile-Butadiene-Styrene (ABS) by fusing NIR and process sensors data, was carried out. Searching for new tools for acquiring and analyzing data in real time is important and therefore I recommend the work for publication.

The authors thank the Reviewer for the positive feedback and judging the work worth of publication as it is.

Round 2

Reviewer 2 Report

The authors made an effort to incorporate their answers as much as they could. The coefficients and the significance of VIF could further be discussed and quantified using classical Fisher statistics or/and ML treatments. This could probably be part of a future paper.